# Adherence to Antibiotic Prescription Guidelines in Four Community Hospitals in Germany

**DOI:** 10.3390/antibiotics13070635

**Published:** 2024-07-10

**Authors:** Joachim Peter Biniek, Frank Schwab, Karolin Graf, Ralf-Peter Vonberg

**Affiliations:** 1Department of Hospital Hygiene, Paracelsus-Hospital am Silbersee, 30851 Langenhagen, Germany; joachim.biniek@pkd.de (J.P.B.);; 2Institute of Hygiene and Environmental Medicine, Charité—Universitätsmedizin Berlin, 12203 Berlin, Germany; 3Institute for Medical Microbiology and Hospital Epidemiology, Hannover Medical School, 30625 Hannover, Germany

**Keywords:** urinary tract infections, bloodstream infections, antibiotic treatment, microbiological diagnostics, intravenous treatment

## Abstract

This retrospective study aimed to assess and compare guideline adherence and treatment costs in the management of urinary tract infections (UTIs) and bloodstream infections (BSIs) in German tertiary hospitals from January 2019 to December 2020. The study analyzed 586 patient records, with 65% diagnosed with UTIs and 35% with BSIs. Antibiotic treatment was given to 98% of patients, but only 65% received microbiological diagnostics. Bacterial growth was observed in 86% of patients with cultures taken, with *Escherichia coli* being the leading pathogen. The treatment was intravenous in 63% of cases, with Ceftriaxone as the leading antibiotic agent. The guideline adherence was found to be low, at 33%. Multivariable logistic regression analysis revealed that patients with urogenital risk factors (OR = 1.589; *p* < 0.001) and increasing age (OR = 1.01; *p* = 0.007) were significantly more likely to receive guideline-concordant treatment for UTIs and BSIs. Additionally, complicating factors such as diabetes and renal dysfunction were associated with higher adherence rates, underscoring the importance of targeted antibiotic stewardship interventions.

## 1. Introduction

The ongoing battle against antimicrobial resistance (AMR) traces back to the discovery of penicillin by Fleming. However, the magnitude of this problem has now reached alarming quantities, warranting global attention. Recognizing the magnitude of the situation, the World Health Organization (WHO) has acknowledged AMR as a global health issue, necessitating the implementation of a comprehensive global action plan [1]. Additionally, a rising global temperature due to climate change will also likely increase bacterial pathogen spread, enlarged usage of antibiotics, and increased AMR in Europe and worldwide [2].

Antibiotic stewardship (ABS) programs aim for a more reasonable and targeted usage of antimicrobial substances to reduce AMR and benefit the patient’s outcome. Numerous studies have consistently demonstrated that adopting this approach holds the potential to enhance patient care quality, mitigate healthcare expenses, minimize the emergence of bacterial resistance, and reduce the risk of adverse events [3]. However, ABS teams with infectious disease specialists may not yet be available in all healthcare facilities. In this case, national and international guidelines on empiric antimicrobial therapy can function as alternative primary source of information on the choice of substance [4].

UTIs are among the most common reasons for a patient to pursue medical help with significantly higher prevalence, especially in women and in the elderly [5]. Consequently, UTIs are among the leading reasons for the onset of antimicrobial therapy in both the outpatient and the inpatient setting [6]. A crucial aspect of effective UTI treatment is knowledge of its causative pathogen. Initial treatment is usually based on an empiric choice as the causative pathogen is yet unknown, while a more focused therapy requires subsequent microbiological diagnostic such as urine cultures (UC). Already hospitalized patients have a high risk of developing hospital acquired UTI associated with a urinary catheter, and data from the “National Wide Point Prevalence” study showed that catheter-associated urinary tract infections (CAUTIs) in 670 instances, with a prevalence rate of 1.04%, being a common cause of nosocomial infection in Germany [7].

BSIs represent a much rarer event but are then associated with high morbidity and mortality [8]. In addition, they also impose a substantial financial burden, with yearly hospital costs of EUR 6.8 billion (USD 7.7 billion) per year in Germany alone [8]. Besides a timely beginning of the empiric administration of antibiotics, microbiological testing by drawing blood cultures (BC) is even more important to facilitate and adjust microbiologically guided antibiotic treatment.

Unfortunately, regimes for UTI and BSI treatment are often sub-optimal or inappropriate despite the availability of official recommendations from experts, and in the case of German community hospitals, there is a notable lack of epidemiological data on the topic of guideline adherence. In the study at hand, we examined the adherence to treatment guidelines and outcomes of UTIs and BSIs in four German community hospitals to further elaborate on this topic.

## 2. Results

### 2.1. Demographic Data

A total of 586 patients were included in the study, while 16 patients were excluded due to insufficient data, and 1 patient was underage. The demographic distribution of the patient cohort is shown in Table A2. Demographic data were from the participating hospitals. Mean age was 78 years (range: 18 to 100). Median age was 82 years (interquartile range (IQR) 71–88), and 355 (60.6%) were female and 231 (39.4%) were male. To ensure comparability among the patient groups from different locations, the length of stay and Charlson comorbidity score (CCS) were used. 

Median length of stay was 6 days (IQR 4–8), with no differences between the locations (*p* = 0.236); median CCS was 1 (IQR 0–3), and it also did not differ between the locations (*p* = 0.088).

### 2.2. Urinary Tract Infections

There were 458 individuals clinically diagnosed with a UTI (ICD-10 codes N39.0, N39.2, N39.3; number of patients with pyelonephritis: 139), and the vast majority (444 patients; 96.9%) of those were confirmed by applying the definitions as described above. A total of 281 pathogens were isolated from urine samples while 339 urine samples remained culture negative. *Escherichia coli* and other coliform bacteria were the main causes of UTIs, as shown in Figure 1. A more detailed description of the pathogen species in urine samples is provided in Table A3 (Appendix A).

### 2.3. Blood Stream Infections

There were 102 individuals clinically diagnosed with an additional UTI-derived (secondary) BSI. A further 81 patients were coded with ICD-10 codes for other types of BSI (A41.X), specifically A41.0 (8 patients), A41.1 (14), A41.2 (2), A41.4 (1), A41.8 (3), A41.9 (25), A41.51 (50), A41.52 (2), and A41.58 (15). Utilizing the assessment approach as mentioned before, BSI was confirmed in 183 of the 586 patients. Overall, 142 bacterial isolates were cultured from 7.070 blood samples. The spectrum of pathogens causing BSI is provided in Figure 2. Once again, *Escherichia coli* and other coliform bacteria were the predominant causative agents. A detailed description of all pathogen species cultured from blood culture bottles is provided in Table A4 (Appendix A).

### 2.4. Use of Antibiotics and Adherence to Guidelines

In 542 (95.4%) of the cases, an antibiotic was prescribed empirically, but only in 61.8% (362) of the cases were samples taken for microbiological culturing and subsequent susceptibility testing. During the study period, 29 different antibiotic agents were utilized (Table A5 and Table A6 (Appendix A)). As shown in Figure 3 for the management of UTIs, the three most frequently used agents were ceftriaxone (63%), amoxicillin/clavulanic acid (22%), and ciprofloxacin (8%). In the treatment of BSIs, the most used agents were ceftriaxone (44%), piperacillin/tazobactam (33%), and amoxicillin/clavulanic acid (23%) (Figure 4). The mean duration of initial antibiotic treatment was 4.51 days (intravenous therapy: 4.8 days; oral formulation: 4.2 days). In 63 (11.6%) of the cases, the treatment was escalated by switching to a broader spectrum antibiotic. However, in 115 (21.2%) of the cases, the treatment could be de-escalated to a narrower spectrum substance. There was adherence to the established guidelines for antimicrobial therapy in less than half (241 (41.1%)) of the cases.

### 2.5. Multivariable Analysis

#### 2.5.1. First-Line Antibiotic Treatment Adherence to Guidelines

Multivariable logistic regression analysis revealed several key factors associated with adherence to first-line antibiotic prescription guidelines (Table 1). Patients with preexisting respiratory conditions were significantly more likely to receive guideline-concordant treatment (OR = 1.52; *p* < 0.020) as well as patients with preexisting gastrointestinal conditions (OR = 1.27; *p* = 0.002), liver and pancreas conditions (OR = 1.46; *p* = 0.002), and psychiatric conditions (OR = 1.46; *p* = 0.002). In addition, an elevation in patient age was linked with a significant increase in guideline adherence (OR = 1.01; *p* = 0.007). However, specifically, female gender was associated with a significantly lower likelihood to receive guideline-concordant treatment (OR = 0.66; *p* = 0.007). Overall, guideline adherence varied significantly among the hospitals.

#### 2.5.2. Antibiotic Treatment Adherence to Guidelines Depending on Microbiological Proof

Results of the multivariable logistic regression analysis for the outcome “antibiotic treatment adherence to guidelines depending on microbiological proof” is shown in Table 2. A total of 334 patients received at least one microbiological proof and stayed ≥3 days in the hospital. In this cohort, preexisting urogenital condition (OR = 1.59, *p* < 0.001), multidrug-resistant organisms tested (OR = 0.50; *p* = 0.006), and parenteral nutrition (OR = 5.26; *p* < 0.001) were significantly associated with the outcome, but not age and gender.

## 3. Discussion

### 3.1. Microbiological Findings UTI

The main causative infectious agents of UTI in our data were *Escherichia coli* (60%), *Enterococcus faecalis* (12%), *Klebsiella pneumoniae* (9%), and *Proteus* spp. (9%). These findings are in accordance with those obtained from various national and international sources, including the Antimicrobial Resistance Epidemiological Survey on Cystitis (ARESC) conducted in nine European countries plus Brazil [9,10]. They are also backed by comprehensive reviews on the topic [11] and clinical trials from the US and Jordan [12,13].

The significant share of the pathogens identified was of *Escherischia coli*, which accounted for 76.7%, followed by *Klebsiella pneumoniae* at 3.5% and *Proteus mirabilis* at 3.4% in the mentioned ARESC Study. Our study also reported *Escherichia coli* to be the predominant pathogen in UTI cases. *Escherichia coli* is the main cause in UTI pathogenesis throughout various geographic regions and healthcare settings. Comparability of pathogen distribution between our study and that of ARESC implies relatively stable global patterns of UTI-causing bacteria, underlining that surveillance for detecting any emerging resistance trend needs to be kept ongoing. The results support a strong base for continuing our targeted antimicrobial stewardship programs, guaranteeing the appropriate use of antibiotics and slowing the development of resistances.

### 3.2. Microbiological Findings BSI

The same applies to the pathogen spectrum for BSIs, as described by others: *Escherichia coli*, *Staphylococcus aureus*, *Klebsiella pneumoniae*, and *Streptococcus pneumoniae* collectively constituted for more than 70% of all community-acquired bloodstream infections as reported by Timsit et al. [14]. In another worldwide study by Diekema et al. performed in 45 countries, the predominant pathogens causing BSIs were *Staphylococcus aureus* (20.7% of cases), *Escherichia coli* (20.5%), *Klebsiella pneumoniae* (7.7%), *Pseudomonas aeruginosa* (5.3%), and *Enterococcus faecalis* (5.2%). Of note, *Staphylococcus aureus* was the predominant pathogen from 1997 to 2004, while *Escherichia coli* emerged as leading cause since 2005. As these were also the main species detected in blood cultures in our study, we consider our data being plausible and valid in principle [15].

### 3.3. Guideline Adherence

Adherence to guidelines and other measures of antibiotic stewardship are well-known tools to improve patient care significantly (reduced toxicity, reduced mortality, reduced selective pressure) [3]. The overall hospital LOS is shortened and the use of second- and third-line treatments are limited, leading to reduced rates of multidrug-resistant organisms, fewer *Clostridioides difficile* infections, and less costs [16,17,18].

Unfortunately, the study at hand from German hospitals shows a tremendous extent of deviation in routine patient care from guidelines for diagnosing and treating UTIs and BSIs in community hospitals. Similar results have been published years prior: An observational study from nine European countries on guideline adherence for diagnosis and treatment of lower UTIs found significant differences between participating countries, showing the best compliance rates for Norway and Denmark (almost 100%), a moderate compliance in the Netherlands (70%), and rather low performance in Germany (40% guideline adherence only) [19]. This might at least partly be due to a higher likelihood of prescribing antimicrobial substances in Germany in general. For example, Dik et al. reported a 10% increase in prescription rates of antibiotics in pediatrics when comparing data from Germany with the Netherlands [20]. A recently published questionnaire on adherence to UTI guidelines among 307 German urologists showed that only 35% of the physicians followed those documents in at least 80% of cases. The main reasons for non-adherence were “personal experience” and lacking practicality of the guidelines in individual cases. Furthermore, 12% of the urologists also considered ignorance as a likely reason for non-compliance [21].

Uncertainties and lack of knowledge with regard especially to UTI guidelines are a well-known problem in Germany: In a theoretical framework, Neugebauer et al. conducted a single-blinded, randomized controlled study involving 166 participants (50% doctors and 50% medical students). The study revealed that only 27% of the subjects could correctly diagnose a UTI and only 20% could initiate an appropriate therapy, despite the availability of aiding documents on the Internet and a Pocket-Guide [22]. Furthermore, the perception and attitude towards antibiotic therapy varies among the population in different countries in general: Zilinskas et al. demonstrated that the general interest in this topic is up to twice as high in countries such as the Netherlands (score: 1.86), Finland (1.76), and Sweden (1.54) compared to Germany (0.80). With respect to the implementation of measures to estimate and encounter the obstacle of development of antimicrobial resistance, Germany was categorized in the third (with the first being best and fourth being worst) cluster of the European Council [23].

Spoorenberg et al. observed a 65% adherence to national and local tailored antibiotic guidelines in 1.252 patients from Dutch university and community hospitals when treating complicated UTIs (pyelonephritis). This is quite like our findings of 41% (adherence to diagnostic and treatment guideline) to 72% (adherence to treatment guideline only). Of note, only the adherence to local guidelines reduced the average length of hospital stay (LOS) by 1.4 days (7.3 days compared to 8.7 days). However, their diagnostic rate was much higher (80% compared to 60% in the study at hand). Furthermore, they changed intravenous to oral therapy in 54% of the cases compared to 19% in our patients. Usually, quick oralization allows for an earlier discharge of patients, and when a timely transition from intravenous to oral antibiotic therapy was feasible within 72 h, they observed a substantial decrease in LOS by 4.3 days (4.8 days versus 9.1 days; *p*< 0.001). On the other hand, compared to our study, one should mention that on average, the Dutch patients were approximately 10 years younger, and their length of stay (LOS) was still twice as long (>8 vs. 4 days) [24].

In a prospective single-center study in a US tertiary hospital, Zatorski et al. reported a 100% diagnostic adherence to UTI guidelines due to a relatively strict surveillance in the study design. Once again, *Escherichia coli* was the main cause of UTIs. However, despite the direct observation of the physicians, treatment adherence of HCWs remained extremely poor due to clinical uncertainties (only 37%), especially when dealing with cystitis cases with a history of prior UTIs and presenting with back pain or abdominal pain only. Often, oral fluoroquinolones were their primary choice of treatment instead of initial intravenous therapy. The approach of oral broad-spectrum therapy may stem from the unique setting of the emergency department, characterized by limited time and patient information, as well as the inability for follow-up [25].

Plate et al. report an 85% guideline adherence from Swiss general practitioners for UTI treatment in the outpatient setting without follow-up visits [26]. As mentioned before regarding the Netherlands, Switzerland also has detailed data on antibiotic consumption as well as epidemiological data on bacterial species and antimicrobial resistance. Furthermore, since 2015, there has been a national campaign in Switzerland called the “National Strategy on Antibiotic Resistance” (StAR) with a catchy slogan (“use wisely, take precisely”) from their Federal Office of Public Health FOPH, suitable for both medical experts and patients [27,28]. Of note, the Swiss UTI guideline was recently updated in March 2023, while up to this day, the German UTI guideline remains officially outdated since 2022.

But there are also some promising findings. Schmiemann et al. evaluated data on 102,000 UTIs from a German medical insurance company for the city of Bremen from 2015 through to 2019 in patients of all ages (main age group: 75 to 84 years). Fluoroquinolones (26%), Fosfomycin (16%), and sulfamethoxazole/trimethoprim (14%) used to be the substances of first choice. Fortunately, there was a trend towards better guideline adherence over time, including a decrease in fluoroquinolone prescription (−20.7% within four years) and an increase in the use of Fosfomycin (+8.8%) and Pivmecillinam (+7.4%) [29].

The cost savings observed with non-guideline-compliant treatment may be attributed to the use of more affordable, generic antibiotics, contrasting with the higher costs associated with guideline-recommended options. However, it is important to acknowledge that this analysis solely focuses on antibiotic expenditures, excluding additional costs such as intravenous preparation and medical instruments. While non-guideline-adherent treatment may appear financially advantageous in the short term, it carries potential risks of compromised patient outcomes, increased antibiotic resistance, and elevated long-term healthcare expenses due to treatment failures and complications. Further investigation is warranted to comprehensively understand the underlying reasons for the observed cost disparities and their implications for patient care and healthcare economics. One potential strategy to promote higher levels of guideline adherence involves implementing regular evaluations in collaboration with the quality management department, including point prevalence studies.

Our analysis revealed that patients with urogenital risk factors were significantly more likely to receive treatment in accordance with clinical guidelines for UTIs (OR = 1.589; *p* < 0.001). These results suggest that healthcare providers are more vigilant in adhering to guidelines for patients with urogenital risk factors, likely due to the increased complexity and potential complications. This aligns with Van Buul et al. [30], who identified six categories of factors influencing antibiotic prescribing: clinical situation, advance care plans, diagnostic resource utilization, perceived risks, influence of others, and environmental factors. The presence of urogenital risk factors likely increases the clinical situation and perceived risks, thus encouraging staff towards stricter adherence to guidelines. A study by Hummers-Pradier et al. [31] showed that patients with risk factors for pyelonephritis like fever and flank pain were more likely to receive antibiotics, which aligns with guideline recommendations, but additional risk factors like impaired renal function or underlying urogenital diseases were not specifically tested.

We observed a slightly though statistically significant increase in the likelihood of guideline adherence with increased age of patients (OR = 1.01, *p* = 0.001). This is in line with the findings of Van der Zande et al. [32] and may be due to the complexities associated with their treatment. Van der Zande et al. also observed that prescribers may avoid using Nitrofurantoin in elderly patients due to its contraindication in cases of impaired renal function, although this contraindication is primarily relevant for those with significantly impaired renal function only. The analysis showed a significant difference between genders. Specifically, females were less likely to adhere to the guidelines compared to males. The odds ratio for females was 0.66 (95% CI: 0.49–0.88, *p* = 0.005), indicating that females had 34% lower odds of adhering to the guidelines compared to males. These findings are in line with the work of Berninghausen et al. [33] who showed in a retrospective study that women are less likely to receive blood cultures when necessary due to sepsis. Furthermore, the study noted that men have a higher incidence of sepsis than women, which might influence decisions to prioritize not only adequate BC sampling but also guideline-adherent therapy for male patients.

## 4. Limitations

Several limitations with respect to our study remain, which shall be addressed in the following.

### 4.1. Heterogeneity of Facilities and Generalization of Results

The demographic data show that the patient populations among the four participating hospitals were comparable in general. There were no significant differences with respect to the distribution of age (*p* = 0.06), gender (*p* = 0.05), and overall guideline adherence (*p* = 0.62). However, there were significant differences regarding the type of infection (*p* < 0.001) and adherence to first-line therapy (*p* < 0.001). This may have been due to the different foci of the four hospitals, including the availability of surgical departments and intensive care units. Generalization of the results should therefore always be carried out with caution and may be limited only to community hospitals that care for an increasing number of older patients with a variety of underlying diseases. For example, patients in tertiary care hospitals such as university hospitals and emergency care facilities tend to be younger on average (55.1 years in 2017) [34], while patients admitted for elective treatment and treatment of acute UTI are mainly in the range of 70 to 79 years in Germany [5,35]. These findings are in line with recently published data from Spain, where most inpatients treated for UTIs were between 75 and 84 years [36]. The incidence of BSIs shows a linear increase after the age of 40 years in adult patients, and in-hospital mortality is highest in the elderly [37]. According to Fleischmann-Struzek et al., most patients with a BSI in Germany are between 60 and 70 years old [37]. On average, due to the types of participating hospitals, our patient population was approximately 10 years older than that. This should be considered when interpreting the results.

### 4.2. Insufficient Diagnostics

Microbiological sampling was performed in only 62% of our cases. And follow-up blood cultures were only rarely taken. Unfortunately, this seems to be a common phenomenon. Chardavoyne et al. also report a rather low rate of urine cultures (306 of 420 patients; 73%) from a US teaching hospital in female patients (2 to 50 years of age) presenting with cystitis and pyelonephritis [12]. These are quite alarming findings as timely knowledge of the species and its susceptibility testing are crucial for the choice of the optimal antimicrobial therapy of UTIs, BSIs, and many other bacterial infections. For example, Bavaro et al. [38] showed in a proof-of-concept study that, among other measures such as procalcitonin guidance and abdominal ultrasound, follow-up blood cultures in Gram-negative BSI might reduce 28 day mortality from 35% to 28%. Thus, there is obviously a need for better education of staff on the diagnostic and therapeutic importance of urine and blood cultures in all participating hospitals. Of note, dependence on external microbiological laboratories may also delay diagnostic findings.

### 4.3. Selection Bias

As mentioned above, the patient population in our study was rather small, rather old, and presented with a large number and variety of comorbidities. The choice of a hospital of paramedics for treatment of emergencies is strongly dependent on the treatment options available in the specific hospital. In our data, a selection bias seems very likely due to the lack of an advanced radiology department (e.g., missing CT), the availability of basic surgical interventions, and limited ICU capacities in some of the participating hospitals. Further studies should address this topic in more detail as there are quite a number of community hospitals in Germany with rather limited medical resources.

### 4.4. Recall Bias

The study at hand has a retrospective design. All results and conclusions rely on medical documentation. We cannot assure that the treatment and the summarized documentation of results were carried out by the same individual. This way, we cannot exclude the possibility of recall bias to some extent.

### 4.5. Information and Observer Bias

Retrospective studies rely on documentation of others in the past. We often noticed a lack of information on clinical data, for example, on patients’ weights and vital signs. The length of antimicrobial therapy was also often missing in the discharge letters of the physicians and had to be retrieved from other data sources. Retrospective interpretation of some type of clinical data (e.g., UTI classification as complicated) may also be biased to some extent. Therefore, we cannot totally exclude minor errors in the raw data set.

## 5. Material and Methods

### 5.1. Design

We performed a retrospective analysis of charts of patients cared for in four German community hospitals (150 to 300 beds each) between January 2019 and December 2020.

### 5.2. Patients

Adult patients diagnosed with an UTI and/or a BSI as defined by the International Classification of Diseases (ICD-10) codes A41 and N39.0 along with the corresponding subcategories were included [39,40]. Details on these subcategories are provided in Table A1 (Appendix A). Patients aged under 18 years, pregnant patients, and patients with incomplete medical records were excluded.

### 5.3. Ethics

Individual consent of the patients was not required due to the retrospective nature of the study without any implications for further treatment and anonymous handling of data. Nevertheless, the approval of the ethics committee of Hannover Medical School was obtained prior to this study [Nr. 9845_BO_K2021].

### 5.4. Demographic Data

We collected the following characteristics:Age;Gender;Duration of inpatient stay;ICD-10 main diagnosis;ICD-10 codes of secondary diagnoses by organ systems;The Charlson comorbidity score [41].

### 5.5. Data on Infections

We checked for

The type of infection (UTI and/or BSI);Its causative pathogen;The clinical outcome of the patient.

For any UTIs code N39.0 (“urinary tract infection, site not specified”), we determined whether there was a cystitis or a pyelonephritis present by examining medical records in detail including description of symptoms, vital signs, physical examinations, and the healthcare provider’s assessment. Patients with fever, chills, flank pain, or tenderness of the costovertebral angle were classified as pyelonephritis [9,42]. In the case of BSI, we used the Sequential Organ Failure Assessment (SOFA) Score [43] to confirm that a BSI was correctly coded; if the SOFA score was not documented, we calculated the quick-SOFA score [43] from the findings in the healthcare provider’s charts instead.

### 5.6. Data on Antimicrobial Therapy

The following data of antibiotics were obtained:Type of substance(s);Timely administration (during visit of the emergency department);Appropriateness of substance in terms of pathogen and type of infection;Dosage, duration, and route administration according to guidelines;Possibility of therapy modification (de-escalation or escalation) according to microbiological findings.

### 5.7. Adherence to Guidelines

We primarily assessed the degree of adherence to national and international treatment guidelines as well as to hospital intern guidelines based on the national guidelines for the treatment of UTIs [9,42] and BSIs [43] in adult patients. We also determined the impact of guideline adherence on the clinical outcome. Secondarily, we examined the variation of microbiological findings between the four participating hospitals.

In the treatment of uncomplicated UTIs, Nitrofurantoin and Fosfomycin were considered as guideline-adherent oral options, and in cases of complicated UTIs, as an oral option, Levofloxacin, Cefpodoxim, or Ceftibuten were considered as guideline adherent. If necessary, parenteral administration of Ceftriaxon, Levofloxacin, or Piperacillin with Tazobactam was considered adequate. In cases of BSIs with a pulmonary focus, Ampicillin with Sulbactam, Piperacillin with Tazobactam, or Meropenem were judged as adequate, depending on the severity of the symptoms, often in combination with Azythromycin.

### 5.8. Statistics

In the descriptive analysis, number and percent or median and interquartile range (IQR) were calculated. Differences were tested by chi-squared test or Wilcoxon rank-sum test and/or Kruskal–Wallis test. To analyze factors associated with the antibiotic treatment adherence to the guidelines, univariable and multivariable logistic regression analyses were performed. Adjusted odds ratios with 95% confidence intervals were calculated by generalized linear models, which consider the cluster effects of hospital. In the multivariable analysis, for the outcome “first-line antibiotic treatment adherence to guidelines”, the following parameters were considered: hospital, age, gender, and preexisting comorbidity conditions. In the model with the outcome “antibiotic treatment adherence to guidelines depending on microbiological proof”, we analyzed all patients with at least microbiological proof and length of stay greater or equal to 3 days. In addition to the parameters mentioned above, the following parameters observed during hospital stay were used for this outcome: length of stay (LOS), intensive care unit (ICU) stay, use of indwelling urinary catheter, multidrug-resistant organisms (MDROs) status taken, MDRO colonization, MDRO infection, blood transfusion, parenteral nutrition, immunosuppression, chemotherapy, and other invasive devices. All mentioned parameters were added to the multivariable model, then non-significant parameters were excluded stepwise backward by the smallest chi-squared value and *p* < 0.05 in the type III score statistic. For epidemiological reasons, the final model for the outcome “antibiotic treatment adherence to guidelines depending on microbiological proof” was additionally adjusted by LOS, gender, and age.

All analyses were exploratory in nature and performed with IBM SPSS statistics version 29.0 (Armonk, New York, NY, USA). Data were collected and securely stored under a two-password and a two-factor-authentication process (Dell Latitude laptop (Dell Technologies Inc., Round Rock, TX, USA), Windows 10 Pro, Version 22H2 (Microsoft Corporation, Redmond, WA, USA), Okta Verify software (Version 9.20.0.2024.625.038) from Okta Inc. (San Francisco, CA, USA), iPhone 11 (Apple Inc., Cupertino, CA, USA) running iOS 17.5.1.).

### 5.9. Microbiologal Diagnostics

All microbiological testing was performed by the laboratory of an assigned university hospital (Hannover Medical School). The microbiological laboratory performing the microbiological diagnostics for the study at hand is accredited according to DIN EN ISO 15189 guidelines [44].

#### 5.9.1. Urine Samples

Midstream urine was collected in a sterile container. Part of the sample was centrifuged for microscopic sediment analysis (bacteria, leukocytes, and erythrocytes). Urine cultures were then quantitatively examined on appropriate culture mediums in Petri dishes using the automated Walk Away Specimen Processor (WSAP®; bioMérieux, Nürtingen, Germany). Identification of so-retrieved uropathogens was performed after 24 h of incubation at 36 °C using matrix-assisted laser desorption/ionization (MALDI)-time of flight (TOF) technology (Schimadzu, Duisburg, Germany). Susceptibility testing and determination of minimal inhibitory concentrations was then performed via micro dilution (VITEK®-2, bioMérieux, Nürtingen, Germany).

#### 5.9.2. Blood Cultures

Blood cultures were drawn aseptically. Blood culture bottles were then incubated under aerobic and anaerobic conditions using automated BD BACTEC® blood culture media systems (Becton Dickinson, Heidelberg, Germany). After detection of microbial growth, bacteria and yeast were then further processed as described above.

#### 5.9.3. Screening for Multidrug-Resistant Organisms

In addition to clinical sampling, sterile swabs were used for patient screening for possible colonization with multidrug-resistant bacteria according to the recommendations of the German National Infection Control Committee (Kommission für Krankenhaushygiene und Infektionsprävention; KRINKO) at the Robert Koch Institute (RKI). Screening for Methicillin-resistant Staphylococcus aureus (MRSA) included swabs from the nose, the throat, and wounds (if applicable). Screening for Vancomycin-resistant enterococci (VRE) and multidrug-resistant Gram-negative (MRGN) rods such as enterobacterales and non-fermenting bacterial species was performed by swabs from the anal region. Swabs were plated on selective culture media in the above-mentioned WASP® automate. Bacterial colonies were then further processed as described above.

## 6. Conclusions

Compared to other countries, there seems to be a need for improvement in adherence to diagnostic and treatment guidelines of UTIs and BSIs, especially in German community hospitals. Therefore, proper ABS must be strengthened. This includes but is not limited to teaching medical students, updating physicians, and educating patients. For this purpose, specific tutorials for the education of staff and application-based decision making may be helpful items.

## Figures and Tables

**Figure 1 antibiotics-13-00635-f001:**
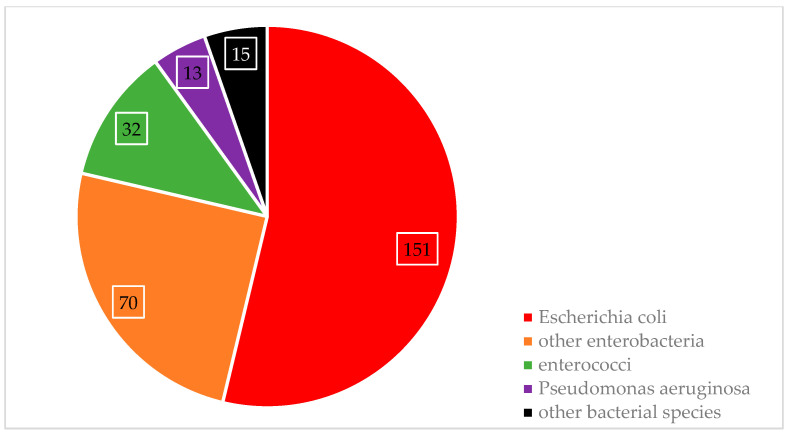
Pathogens causing urinary tract infections.

**Figure 2 antibiotics-13-00635-f002:**
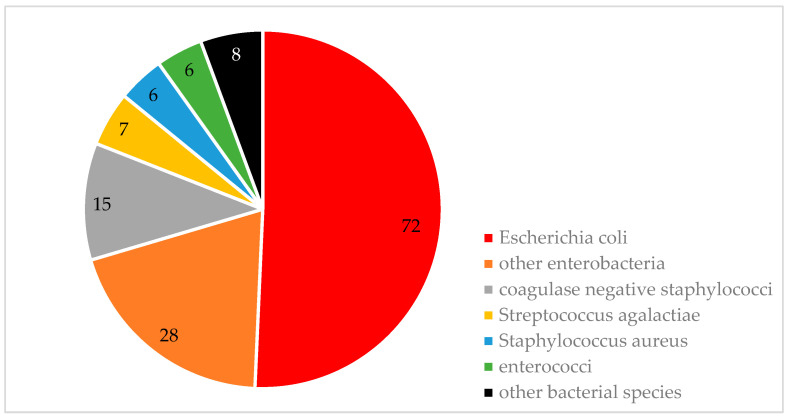
Pathogens causing bloodstream infections.

**Figure 3 antibiotics-13-00635-f003:**
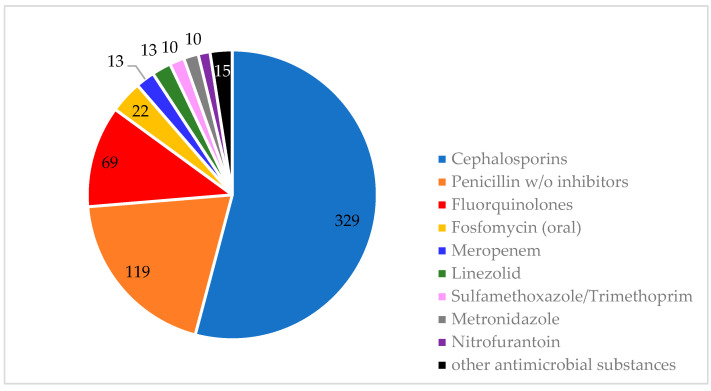
Antibiotics used for the treatment of urinary tract infections.

**Figure 4 antibiotics-13-00635-f004:**
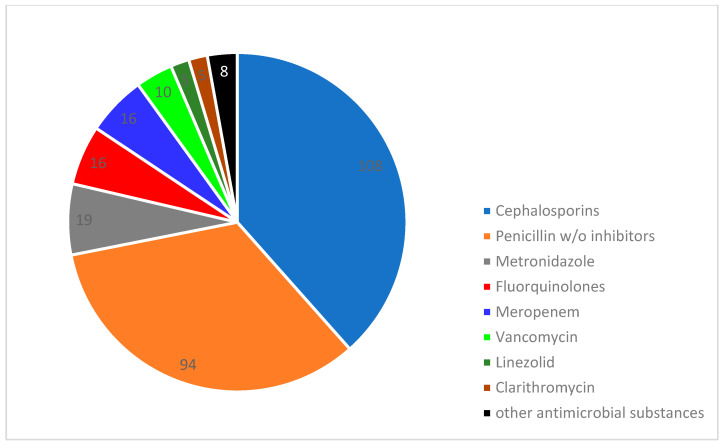
Antibiotics used for the treatment of bloodstream infections.

**Table 1 antibiotics-13-00635-t001:** Results of multivariable logistic regression analysis for the outcome “first-line antibiotic treatment adherence to guidelines” (n = 586 patients).

Parameter	Odds Ratio	95%CI	*p*-Value Type I	*p*-Value Type III
(Intercept)	0.27	0.14–0.53	<0.001	<0.001
Hospital 4	0.97	0.95–0.99	0.001	<0.001
Hospital 3	1.14	1.09–1.2	<0.001	
Hospital 2	0.57	0.54–0.6	<0.001	
Hospital 1	1 = ref			
Age	1.01	1–1.02	0.007	0.007
Female	0.66	0.49–0.88	0.005	0.005
Male	1 = ref			
Preexisting respiratory conditions (n = 161)	1.52	1.07–2.16	0.020	0.020
Preexisting gastrointestinal conditions (n = 232)	1.27	1.04–1.54	0.017	0.017
Preexisting liver and pancreas conditions (n = 147)	1.46	1.15–1.84	0.002	0.002
Preexisting psychiatric conditions (n = 247)	1.19	1–1.42	0.051	0.051

CI, confidence interval.

**Table 2 antibiotics-13-00635-t002:** Results of multivariable logistic regression analysis for the outcome “antibiotic treatment adherence to guidelines depending on microbiological proof” (n = 334 patients with length of stay ≥3 days and at least one microbiological proof).

Parameter	Odds Ratio	95%CI	*p*-Value Type I	*p*-Value Type III
Model without age and sex				
(Intercept)	0.22	0.17–0.29	<0.001	0.492
Hospital 4	5.49	4.59–6.57	<0.001	<0.001
Hospital 3	5.04	4.12–6.18	<0.001	
Hospital 2	1.47	1.28–1.68	<0.001	
Hospital 1	1 = ref			
Preexisting urogenital condition	1.59	1.35–1.87	<0.001	<0.001
Multidrug-resistant organisms (MDROs) tested	0.50	0.31–0.83	0.006	0.006
Parenteral nutrition	5.26	2.64–10.46	<0.001	<0.001
Model with age, sex and length of stay				
(Intercept)	0.35	0.15–0.83	<0.017	0.019
Hospital 4	5.10	3.91–6.65	<0.001	<0.001
Hospital 3	5.22	4.53–6.00	<0.001	
Hospital 2	1.51	1.35–1.70	<0.001	
Hospital 1	1 = ref			
Age	1.00.	0.99–1.01	0.671	0.671
Female	1.02	0.79–1.31	0.871	0.871
Male	1 = ref			
Length of stay	0.96	0.89–1.03	0.215	0.215
Preexisting urogenital condition	1.64	1.48–1.82	<0.001	<0.001
MDRO status taken	0.54	0.30–0.97	0.038	0.038
Parenteral nutrition	9.76	4.87–19.42	<0.001	<0.001

## Data Availability

The data presented in this study are available in article.

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
