# Peer review of "Adherence to Antibiotic Prescription Guidelines in Four Community Hospitals in Germany"

_antibiotics, 2024, doi:10.3390/antibiotics13070635_

Round 1
Reviewer 1 Report
Comments and Suggestions for Authors
In this study, the authors analyzed and determined the guideline adherence associated with urinary tract infections and bloodstream infections in German tertiary hospitals. Please find my comments below:
Abstract: Please provide more details (more 1-2 sentences) in the abstract for your methodology.
Line 14: Full form of UTI and BSI?
Line 17: Please use a different word instead of “Germ”, e.g., bacteria or others.
Line 17: “E. coli” should be “Escherichia coli”. Please use the full form first and then the abbreviation. Also, any organisms’ scientific names should be in italics. Please correct it throughout the manuscript if needed.
Line 20-21: “p” from “p-value” should be in italics. Please correct it throughout the manuscript if needed.
Introduction: Any knowledge gap in your study? If so, please mention it before the objective.
Line 75: Please mention the age of underage here, e.g., <x years..
Table S1: Please provide the reference for this information. Also, you should provide some information for scientific words, e.g., how they were sure that those organisms were MRSA or vancomycin-resistant, and others.
Line 126-127: Please provide the details of “IBM SPSS statistics”, e.g., city and country name with the company name.
Line 123: You performed multivariate logistic regression analyses, which I appreciate. However, you could describe this statistical analysis. How did you perform multivariate logistic regression analyses? How did you create a logistic regression model? How did you select variables to create the regression model? Did you perform a collinearity test or not? If yes, how? Please mention them.
Line 136-137: Have grammatical issues.
Line 143: 281 should be Two-hundred eighty-one.
Line 144-145: Please provide detailed information on how the organisms were isolated.
Line 146: Table S2 should be Table S3 and S4.
Line 167: Tables S4 and S5 should be Tables S5 and S6.
Line 215: I was just wondering if the authors used a systematic review to compare their study’s outcomes. Please clarify this.
Line 216-217: Please discuss this result for the pathogens. You need to discuss them, not just provide the results.
Author Response
Thank you very much for your thorough review and valuable suggestions. We appreciate the time and effort you invested in revising our manuscript, which has greatly contributed to improving the quality and clarity of our work. We have revised the manuscript accordingly and addressed all comments, as detailed below
Abstract: Please provide more details (more 1-2 sentences) in the abstract for your methodology.
-
- We appreciate your suggestion. Additional details have been included in the methodology section of the abstract in lines 15-17 to enhance clarity.
Line 14: Full form of UTI and BSI?
-
- Thank you for pointing this out. We have corrected the issue by providing the full forms as "Urinary Tract Infection (UTI)" and "Bloodstream Infection (BSI)."
Line 17: Please use a different word instead of “Germ”, e.g., bacteria or others.
-
- We have replaced the term "germ" with "bacteria" in line 17 to improve accuracy.
Line 17: “E. coli” should be “Escherichia coli”. Please use the full form first and then the abbreviation. Also, any organisms’ scientific names should be in italics. Please correct it throughout the manuscript if needed.
-
- We appreciate your attention to detail. The full form "Escherichia coli" is now used first, followed by the abbreviation "E. coli" where appropriate. All scientific names have been italicized throughout the manuscript.
Line 20-21: “p” from “p-value” should be in italics. Please correct it throughout the manuscript if needed.
-
- We have italicized the "p" in "p-value" throughout the manuscript as recommended.
Introduction: Any knowledge gap in your study? If so, please mention it before the objective.
-
- Thank you for your suggestion. We have addressed this in lines 62-67, highlighting the notable lack of data regarding guideline adherence in German community hospitals as the main knowledge gap.
Line 75: Please mention the age of underage here, e.g., <x years..
-
- This issue has been corrected in line 76 by specifying the age of underage patients.
Table S1: Please provide the reference for this information. Also, you should provide some information for scientific words, e.g., how they were sure that those organisms were MRSA or vancomycin-resistant, and others.
-
- A reference has been added, and we have addressed the microbiological methodology in sections 2.9-2.9.3, lines 152-182. Additionally, the references have been updated at the end. Thank you for highlighting this.
Line 126-127: Please provide the details of “IBM SPSS statistics”, e.g., city and country name with the company name.
-
- We have provided the necessary details: "IBM SPSS Statistics version 29.0 (Armonk, New York, United States)" in lines 149-150.
Line 123: You performed multivariate logistic regression analyses, which I appreciate. However, you could describe this statistical analysis. How did you perform multivariate logistic regression analyses? How did you create a logistic regression model? How did you select variables to create the regression model? Did you perform a collinearity test or not? If yes, how? Please mention them.
-
- We appreciate your positive feedback on our statistical analysis. Detailed information on the multivariate logistic regression analyses, including model creation, variable selection, has been added in lines 128-147.
A collinearity test was not performed.
- We appreciate your positive feedback on our statistical analysis. Detailed information on the multivariate logistic regression analyses, including model creation, variable selection, has been added in lines 128-147.
Line 136-137: Have grammatical issues.
-
- The grammatical issues in the specified lines 136-137 have been corrected. Thank you for your attention to this matter.
Line 143: 281 should be Two-hundred eighty-one.
-
- This issue has been corrected into Two-hundred eighty-one in line 143.
Line 144-145: Please provide detailed information on how the organisms were isolated.
-
- We have added sections 2.9-2.9.3 to further describe our method for isolating organisms in lines 152-182.
Line 146: Table S2 should be Table S3 and S4.
-
- The tables are now in the correct order. We appreciate your careful review.
Line 167: Tables S4 and S5 should be Tables S5 and S6.
-
- The tables are now in the correct order. We appreciate your careful review.
Line 215: I was just wondering if the authors used a systematic review to compare their study’s outcomes. Please clarify this.
-
- We corrected our fault in line 215; the review was comprehensive, not systematic. Thank you very much for your keen observation and meticulous review.
Line 216-217: Please discuss this result for the pathogens. You need to discuss them, not just provide the results.
- This issue has been corrected. We have expanded the discussion of the results for the pathogens in lines 273-282 to provide a more thorough analysis. Thank you for your valuable input.
The revised manuscript is added.
Thank you again for your thoughtful and supportive review.
Sincerely,
Joachim Biniek
Dr. Frank Schwab
PD Dr. Karolin Graf
Prof. Dr. Ralf-Peter Vonberg
Reviewer 2 Report
Comments and Suggestions for Authors This is a study about adherence to antibiotic prescription guidelines in four community hospitals in Germany (with 150 to 300 beds each) to analyze the ongoing battle against antimicrobial resistance between January 2019 and December 2020. Two types of adult patients were studied, some with urinary tract infections (UTIs) and others with blood stream infections (BSIs). The methodology is widely described. 586 patients were included in the study, women and men between 18 and 100 years old. For the study, demographic and infection data were considered. The data obtained was studied statistically with a very good confidence interval and is clearly presented with very illustrative images with a very well-structured format.The discussion is broad and compelling. It is concluded that an improvement in adherence to diagnosis and treatment guidelines of UTI and BSI is advisable that includes both medical students and the updating of physicians and educated patients. It is an excellent work in which the results obtained are analyzed in detail. The references are adequate and up to date. It is a work worthy of publication.
Author Response
This is a study about adherence to antibiotic prescription guidelines in four community hospitals in Germany (with 150 to 300 beds each) to analyze the ongoing battle against antimicrobial resistance between January 2019 and December 2020. Two types of adult patients were studied, some with urinary tract infections (UTIs) and others with blood stream infections (BSIs). The methodology is widely described. 586 patients were included in the study, women and men between 18 and 100 years old. For the study, demographic and infection data were considered. The data obtained was studied statistically with a very good confidence interval and is clearly presented with very illustrative images with a very well-structured format.
The discussion is broad and compelling. It is concluded that an improvement in adherence to diagnosis and treatment guidelines of UTI and BSI is advisable that includes both medical students and the updating of physicians and educated patients. It is an excellent work in which the results obtained are analyzed in detail. The references are adequate and up to date. It is a work worthy of publication.
-
Thank you very much for your thorough and positive evaluation of our study on adherence to antibiotic prescription guidelines in four community hospitals in Germany. We appreciate your recognition of the comprehensive methodology, the statistical rigor, and the clarity with which we have presented our findings.
Your acknowledgment of the broad and compelling discussion, as well as the relevance of our conclusions regarding the improvement of guideline adherence for UTIs and BSIs, is highly appreciated. We are also grateful for your commendation of our effort to include medical students, updating physicians, and educating patients as part of our recommendations.
We are pleased to hear that you find our work to be worthy of publication. Your encouraging feedback motivates us to continue our research in this critical area of antimicrobial resistance.
Thank you again for your thoughtful and supportive review.
Sincerely,
Joachim Biniek
Dr. Frank Schwab
PD Dr. Karolin Graf
Prof. Dr. Ralf-Peter Vonberg
Reviewer 3 Report
Comments and Suggestions for Authors
This retrospective study tried to compare the guideline adherence in the management of urinary tract infections and bloodstream infections in four different hospitals in Germany.
-
- Line 69 Please check the word order and meaning of the sentence.
- Line 135 „355 (60.6%) female“ What does it mean? 355 were female? Please check.
- Line 144 Escherichia coli and other bacteria should be in italic. Please check the whole text for the same mistake.
- Line 204-5 significantly instead of significant.
- Line 212-5 Please rephrase the sentence to be more meaningful
- Line 216 Please change „In this...“ to „In the mentioned ARESC...“
- Line 420 Why authors didn’t incorporate Table with demographics into the Results section? I suggest to do so. It is hard to follow the main idea of the study.
- Line 422 and Line 424 Add the info about total number of urine cultures in the table caption.
- There is no mentioning in the Methods section what are the criteria for the adherence to the established guidelines. In other words, there is no info about the appropriate antibiotic choice for UTIs and BSIs.
- Line 197 How many cases of preexisting conditions were identified?
- Line 205 What about length of stay?
- Line 207 How many cases of multidrug resistant organisms were identified? What were the criteria to classify the bacteria as a multidrug resistant organism?
- The authors said in the abstract that they compared treatment costs but I haven’t found the info.
- Line 115 and Line 452-5 Mentioned guideline is outdated because EMA issued a final decision on quinolones and fluoroquinolones on 15 November 2018 which affected most of the guidelines.
- Line 115 and Line 456-9 Is it a national or an international guideline used for the adherence testing? It is hard to believe that in Germany an outdated (2011) and international guideline is used by clinicians officially.
- Besides many limitations of the study already stated by the authors two previously mentioned are the main limitation of the study.
Author Response
Thank you very much for your thorough review and valuable suggestions. We appreciate the time and effort you invested in revising our manuscript, which has greatly contributed to improving the quality and clarity of our work. We have revised the manuscript accordingly and addressed all comments, as detailed below:
Line 69 Please check the word order and meaning of the sentence.
-
- This issue has been corrected. Thank you for your attention to this matter.
Line 135 „355 (60.6%) female“ What does it mean? 355 were female? Please check.
-
- Yes, 355 were female. We have clarified this in line 189.
Line 144 Escherichia coli and other bacteria should be in italic. Please check the whole text for the same mistake.
-
- The mistake has been corrected throughout the text.
Line 204-5 significantly instead of significant.
-
- This issue has been corrected to significantly in line 256-257
Line 212-5 Please rephrase the sentence to be more meaningful
-
- This issue has been corrected in line 263-269.
Line 216 Please change „In this...“ to „In the mentioned ARESC...“
-
- Done, in line 272.
Line 420 Why authors didn’t incorporate Table with demographics into the Results section? I suggest to do so. It is hard to follow the main idea of the study.
-
- Good point. We will incorporate the table with demographics into the Results section, of it aligns with the editor's guidelines. Thank you for your helpful suggestion.
Line 422 and Line 424 Add the info about total number of urine cultures in the table caption.
-
- Unfortunately, this data is not retrospectively available.
There is no mentioning in the Methods section what are the criteria for the adherence to the established guidelines. In other words, there is no info about the appropriate antibiotic choice for UTIs and BSIs.
-
- The mentioned topic is now addressed in point 2.7, lines 119-125.
Line 197 How many cases of preexisting conditions were identified?
-
- The absolute cases have been added to Table 1, line 249:
- Preexisting respiratory conditions (n=161)
- Preexisting gastrointestinal conditions (n=232)
- Preexisting liver and pancreas conditions (n=147)
- Preexisting psychiatric conditions (n=247)
- The absolute cases have been added to Table 1, line 249:
Line 205 What about length of stay?
-
- LOS was not significant associated with "Antibiotics treatment adherence to guidelines depending on microbiological proof" (Table 2, line 259, Model with age, sex and length of stay).
Line 207 How many cases of multidrug resistant organisms were identified? What were the criteria to classify the bacteria as a multidrug resistant organism?
-
- The information has been added to Table 2:
- Colonization: 46 cases
- Infection: 27 cases
- Information about the definition and identification has been added to sections 2.9-2.9.3 in lines 151-181.
- The information has been added to Table 2:
The authors said in the abstract that they compared treatment costs but I haven’t found the info.
-
- We apologize for the confusion caused by an older abstract. We have decided to remove the aspects of cost comparison from the study. The focus of the study was not centered on this aspect, and due to its lack of significant impact on the therapeutic approach and the insufficient data available, further investigation in this area was not continued.
Line 115 and Line 452-5 Mentioned guideline is outdated because EMA issued a final decision on quinolones and fluoroquinolones on 15 November 2018 which affected most of the guidelines.
-
- The guideline was still considered up-to-date until 2020. We ruled quinolones and fluoroquinolones as non-guideline adherent based on the EMA statement and the German equivalent "Rote Hand Brief."
Line 115 and Line 456-9 Is it a national or an international guideline used for the adherence testing? It is hard to believe that in Germany an outdated (2011) and international guideline is used by clinicians officially.
-
- Thank you for your insightful comment. In our study, we utilized national guidelines that were considered up-to-date until 2020. Although the referenced guideline dates back to 2011, it was still in official use during the study period, supplemented by updates from national health authorities such as the EMA statement and the German "Rote Hand Brief," which provided additional directives on the use of quinolones and fluoroquinolones. These updates were crucial in guiding clinical practice and ensuring the adherence criteria reflected the most current recommendations.
Thank you again for your thoughtful and supportive review.
Sincerely,
Joachim Biniek
Dr. Frank Schwab
PD Dr. Karolin Graf
Prof. Dr. Ralf-Peter Vonberg
Round 2
Reviewer 1 Report
Comments and Suggestions for Authors
The authors addressed my comments.
Reviewer 3 Report
Comments and Suggestions for Authors
There are no further comments and suggestions.